# Diagnostic Blood Biomarkers for Acute Pulmonary Embolism: A Systematic Review

**DOI:** 10.3390/diagnostics13132301

**Published:** 2023-07-06

**Authors:** Vårin Eiriksdatter Wikan, Birgitte Gladsø Tøndel, Vânia Maris Morelli, Ellen Elisabeth Brodin, Sigrid Kufaas Brækkan, John-Bjarne Hansen

**Affiliations:** 1Thrombosis Research Group (TREC), Department of Clinical Medicine, UiT—The Arctic University of Norway, N-9037 Tromsø, Norway; 2Thrombosis Research Center (TREC), Division of Internal Medicine, University Hospital of North Norway, N-9038 Tromsø, Norway; 3Hematological Research Group, Division of Medicine, Akershus University Hospital, N-1478 Lørenskog, Norway

**Keywords:** pulmonary embolism, biomarker, diagnostic accuracy

## Abstract

(1) Background: The current diagnostic algorithm for acute pulmonary embolism (PE) is associated with the overuse of CT pulmonary angiography (CTPA). An additional highly specific blood test could potentially lower the proportion of patients with suspected PE that require CTPA. The aim was to summarize the literature on the diagnostic performance of biomarkers of patients admitted to an emergency department with suspected acute PE. (2) Methods: Medline and Embase databases were searched from 1995 to the present. The study selection process, data extraction, and risk of bias assessment were conducted by two reviewers. Eligibility criteria accepted all blood biomarkers except D-dimer, and CTPA was used as the reference standard. Qualitative data synthesis was performed. (3) Results: Of the 8448 identified records, only 6 were included. Eight blood biomarkers were identified, of which, three were investigated in two separate studies. Red distribution width and mean platelet volume were reported to have a specificity of ≥ 90% in one study, although these findings were not confirmed by other studies. The majority of the studies contained a high risk of selection bias. (4) Conclusions: The modest findings and the uncertain validity of the included studies suggest that none of the biomarkers identified in this systematic review have the potential to improve the current diagnostic algorithm for acute PE by reducing the overuse of CTPA.

## 1. Introduction

Acute pulmonary embolism (PE) is a frequent and potentially fatal disease with an annual incidence of 60 to 80 cases per 100,000 inhabitants and a 1-year mortality rate of approximately 20% [1,2,3,4]. Diagnosing PE is a clinical challenge, and the diagnosis is frequently missed or delayed due to unspecific signs and symptoms [5,6,7]. Current guidelines, including the European Society of Cardiology (ESC) 2019 guidelines and the American Society of Hematology (ASH) 2018 guidelines, recommend using a diagnostic strategy consisting of a clinical pretest probability assessment, D-dimer assay, and radiological imaging (mainly CTPA) [8,9]. In patients with a low or intermediate clinical probability of PE and a negative D-dimer test, acute PE can be ruled out because of the excellent negative predictive value of D-dimer [8]. In patients with a high clinical probability of PE or positive D-dimer, CTPA is used to confirm or rule out the diagnosis with a high positive predictive value and specificity [8,9].

Although this diagnostic algorithm is validated as being safe and efficient [8,9], recent studies have reported the overuse of CTPA in clinical practice. Pulmonary embolism is found in only 8–25% of all patients referred for CPTA [10,11,12,13], and diagnostic studies with consecutive patient sampling report that 32–55% of all CTPAs are “unnecessary” since they are performed on patients with low pretest probabilities and/or a negative D-dimer [14,15,16]. Regardless of the underlying causes of these observations, the low proportion of confirmed PE in patients undergoing CTPA, even when the clinical algorithm is optimally applied, emphasizes a need for further improvement in the current algorithm. A blood biomarker that can be combined with the D-dimer and used to rule out PE in a larger proportion of patients with suspected PE (without the need for further referral for CTPA) would be desirable. A reduction in the use of CTPA would benefit both patients and the healthcare system since it would reduce exposure to radiation and contrast-induced morbidity, and also lower the use of healthcare resources and costs [8,17,18,19].

An overview of previous studies on diagnostic blood biomarkers of acute PE (other than D-dimer) can be useful to identify knowledge gaps and guide future research in this field. Therefore, the aim of this systematic review was to summarize the available literature on the diagnostic performance of other blood biomarkers of acute PE than D-dimer in outpatients or patients admitted to an emergency department with suspected acute PE.

## 2. Materials and Methods

This systematic review is reported in accordance with the preferred reporting items for systematic reviews and meta-analyses of diagnostic test accuracy studies (PRISMA-DTA) [20]. The prespecified protocol for this study was published 7 February 2022 on PROSPERO (registration number: CRD42022301515) available at https://www.crd.york.ac.uk/prospero, accessed on 19 January 2023.

### 2.1. Data Sources and Search Strategy

We systematically searched the Medline and Embase electronic databases (via the OvidSP platform) on the 18 January 2023. The search strategy was based on a combination of subject heading terms and free text terms related to acute PE and blood biomarkers. The complete search strategies and search terms are available in Appendix A. The search was limited to studies on humans, published in English, from the 1 January 1995 and onwards. In addition, a restriction on publication type was applied in Embase to exclude conference abstracts from the search, and the focus function was applied to the subject heading “lung embolism” to limit the number of retrieved records. We also performed a manual search through the reference lists of relevant retrieved articles and reviews to identify potential studies that were not captured by the original search.

### 2.2. Study Selection and Eligibility Criteria

To identify eligible studies, three teams, each consisting of two independent reviewers, screened the titles and abstracts of all retrieved records, and subsequently, assessed full-text articles. Thus, the study selection was conducted independently and in duplicate. A third reviewer (JBH or SKB) resolved discrepancies when necessary. A standardized, piloted screening form was used to facilitate the study selection process.

Cross-sectional studies, cohort studies, or randomized controlled trials that reported data on diagnostic test accuracy (i.e., sufficient data to determine sensitivity and specificity) of blood biomarkers in persons with suspected acute PE were eligible for inclusion in this systematic review. Diagnostic case–control studies using healthy controls (two-/multi-gate design) were excluded since such studies generally are known to overestimate the index test’s accuracy compared to clinical practice [21].

To be included, studies had to enroll adult patients (≥18 years) with suspected first or recurrent acute PE that had been admitted to an emergency department or an outpatient setting. Studies with a population consisting of a mixture of in- and outpatients or predominantly cancer or pregnant patients were excluded. Further, studies had to report the test accuracy of at least one diagnostic blood biomarker for acute PE, measured within 24 h of admission. Studies solely reporting diagnostic accuracy data of D-dimer were excluded.

Studies using CTPA as the reference standard were eligible for inclusion, in line with the ESC 2019 recommendation of using CTPA as the method of choice for visualizing pulmonary vasculature, due to its availability in clinical practice and low rate of inconclusive results [8]. We accepted deviation from CTPA as the reference standard if the proportion of patients who had another reference standard test performed (e.g., V/Q-scan or autopsy) was less than 10% of the total study population, and the choice was justified. Death, severe renal failure, iodine allergy, pregnancy, breastfeeding, or hyperthyroidism were considered reasonable explanations for choosing another reference standard other than CTPA [8]. Studies conducted before the 1st of January 1995 were excluded since CTPA was not a commonly used diagnostic procedure for acute PE until the late 1990s in both North America and Europe [22,23].

Only full-text articles and letters to the editor were assessed for eligibility. If more than one record was reported on the same results of an index test in the same study population, we included only the first publication. For studies where the eligibility was uncertain, as well as studies that did not report sufficient data to calculate the diagnostic accuracy of the index test(s), the corresponding author was contacted through email for clarification and further information.

### 2.3. Data Extraction

Data extraction was carried out by two reviewers (VEW and BGT), independently and in duplicate. A standardized, piloted form was used to extract data on general study characteristics (including authors, year of publication, country, study type, and setting), study population characteristics (e.g., eligible participants, exclusion criteria, total study size, mean age, and gender distribution), index test(s) (including timing, measurement technique, execution, interpretation, and threshold), and reference standard (including criteria for a positive test result, timing, execution, and interpretation). In addition, we extracted accuracy data to create 2 × 2 contingency tables (i.e., true positives (TPs), true negatives (TNs), false positives (FPs), and false negatives (FNs), as reported by the studies, or we calculated these data from reported estimates of diagnostic accuracy (i.e., sensitivity and specificity), and disease prevalence).

### 2.4. Risk of Bias Assessments of Included Studies

To facilitate risk of bias assessment in each individual study, we used the quality assessment of diagnostic accuracy studies-2 (QUADAS-2) revised tool, which consists of four bias domains: patient selection, index test, reference standard, and patient flow and timing [24]. Two reviewers (VEW and BGT) evaluated all included studies, independently and in duplicate. Final judgments were made in consensus with the involvement of a third reviewer (JBH or SKB), when necessary.

### 2.5. Data Synthesis

For qualitative data synthesis, study characteristics (including details of the patient population), index test(s) with diagnostic thresholds, and reference standards, as well as the risk of bias assessments, were presented in both tabular and narrative formats. To reproduce reported estimates of sensitivity and specificity for each index test with corresponding 95% confidence intervals (95% CIs), along with other estimates of diagnostic accuracy, we employed extracted or calculated 2 × 2 contingency table accuracy data into Review Manager (RevMan5 Version 5.4, The Cochrane Collaboration, 2020). The results of each study were presented numerically and graphically with paired forest plots. In the review protocol, we prespecified that we would require at least five studies on the same index test to perform any quantitative analysis. The rationale for this requirement was that meta-analyses based on minimal studies are considered to have a limited clinical value [25]. In addition, as diagnostic test accuracy (DTA) studies (i) are prone to a high degree of heterogeneity, (ii) lack good heterogeneity tests, and (iii) have two dependent summary statistics, random effects meta-analyses models are recommended for a systematic review of DTA studies. These models are often troublesome to converge with sparse data [25].

## 3. Results

In total 6474 non-duplicate records were identified in our systematic review search of electronic databases and manual review of reference lists. Seventy-eight studies were included for full-text assessment, of which, six studies [26,27,28,29,30,31] fulfilled all the eligibility criteria and were included in the systematic review (see Figure 1 for PRISMA flowchart). At the stage of full-text assessment and data extraction, reasons for exclusion were two-gate case–control study design (n = 32), ineligible study population (n = 7), the study authors did not aim to assess sensitivity and specificity or did not report sufficient data to estimate these measures (insufficient reporting) (n = 16), unacceptable reference standard (n = 12), disease of interest was not acute PE (n = 3), the diagnostic test results for the same index test on the same study population was reported 2 years earlier by another journal (n = 1), and/or ineligible publication type (n = 1). An overview of excluded studies along with the main reasons for exclusion is provided in Appendix A.

### 3.1. Risk of Bias Assessment

An overview of the individual risk of bias assessments of all included studies is provided in Figure 2. The review-specific QUADAS-2 form used for risk of bias assessment is presented in Appendix A. The risk of bias in all six studies was deemed to be high in at least one (bias) domain. In five studies [26,27,28,30,31], the risk of bias arising from patient selection was considered high because the studies excluded patients diagnosed with common differential diagnoses of acute PE. In contrast, such inappropriate exclusions were not performed in the study by Flores et al. [29], and thus, the study was considered to have a low risk of bias in this domain. An overview of the exclusion criteria applied in each study is provided in Appendix A.

The conduct and interpretation of the index tests were, in general, well described, and five studies [26,27,29,30,31] were assessed to have low risks of bias in this domain, while only Ebrahimi et al. [28] had an unclear risk of bias in this domain. On the other hand, inadequate reporting of the conduct and interpretation of the reference standard was a major issue across the included studies, and the risk of bias in the reference standard domain was, therefore, deemed unclear in five studies [26,27,29,30,31]. Moreover, the unusual algorithm of acute PE validation in the study by Flores et al. [29], including combinations of several reference standards, was considered to increase the risk of bias in the domain.

Insufficient reporting on patient flow and/or time intervals between the index test, reference standard, and other interventions were issues of concern in four studies [26,27,28,31]. These studies did not provide a flow diagram to illustrate the patient flow throughout the study, nor did they clearly report the timing of the conduct of the index test and reference standard, and whether any treatment was withheld until the tests were performed. Consequently, the four studies were considered to have unclear risks of bias in this domain [26,27,28,31]. In contrast, the patient flow, and the time sequence of interventions in the study by Flores et al. [29] were well described narratively and by a flowchart, meaning they were deemed to have a low risk of bias in this domain. The study by Huang et al. [30] was judged to have a high risk of bias since we were unable to reproduce the same reported diagnostic accuracy estimates based on their 2 × 2 contingency table.

### 3.2. Study Characteristics and Findings

Study characteristics of the included studies, as well as index tests and reference standards, are reported in Table 1. Estimates of diagnostic accuracy for a total of eight different blood biomarkers for acute PE were reported by the studies. Sensitivity and specificity with 95% CIs are presented numerically and by paired forest plots in Table 2 and Figure 3, respectively. Other diagnostic accuracy estimates, including positive predictive value (PPV), negative predictive value (PPV), and overall percent agreement, are presented in Table 2. As there were only a maximum of two studies reporting accuracy data on the same index test, our prespecified requirements for performing a meta-analysis were not met; thus, we did not calculate pooled estimates of sensitivity and specificity, or create any summary receiver operating curves.

In the study by Celik et al. [26], 248 patients with suspected acute PE admitted to a hospital in Turkey were included. Blood samples were obtained within 2 h of admission and an automatic blood counter (A Sysmex XE-2100; Symex, Kobe, Japan) was used to determine red cell distribution width (RDW). All participants underwent CTPA as the reference standard, and the acute PE prevalence was 45%. The diagnostic data were collected retrospectively. An optimal RDW threshold of <18.9% was calculated and a sensitivity of 21% achieved was achieved (95% CI, of 14% to 29%), with a specificity of 93% (95% CI: 88% to 97%) for acute PE diagnosis.

Çevik et al. [27] included 128 patients with suspected acute PE admitted to an emergency department at a Turkish hospital. An electronic cell counter (Beckman Coulter LH 780) was used to determine mean platelet volume (MPV), platelet distribution width (PDW), and platelet count in blood samples drawn at admission as a part of the routine blood count. The acute PE diagnosis was confirmed by using CTPA as the reference standard, and the acute PE prevalence was 48%. The test results from the platelet indices and CTPA were collected retrospectively. For MPV, a threshold of ≤9 fL achieved the highest sensitivity of 34% (95% CI: 23% to 48%) and specificity of 90% (95% CI: 80% to 96%). For PDW, a threshold of >12.8 fL achieved the highest sensitivity of 62% (95% CI: 49% to 74%) and specificity of 72% (95% CI: 59% to 82%). For platelet count, a threshold of ≤254 × 10^3^ platelets per µL achieved the highest sensitivity of 62% (95% CI: 49% to 74%) and specificity of 57% (95% CI: 44% to 69%).

Ebrahimi et al. [28] prospectively recruited 267 patients non-randomly with suspected acute PE referred to an emergency department at an Iranian hospital. Patients with shortness of breath and chest pain in addition to a negative D-dimer test were eligible. The investigated index tests N-terminal prohormone of brain natriuretic peptide (NT-proBNP) and troponin I were analyzed using a CLIA Kit for Humans (Cloud clone, USA) and quantitative enzyme-linked immunosorbent assay (ELISA) test kit (Diagnostic Automation Inc., USA), respectively. All patients received CTPA as a reference standard, and the prevalence of acute PE was 45%. The calculated threshold of >100 pg/mL for NT-proBNP achieved the highest sensitivity of 85% (95% CI: 78% to 91%) and specificity of 80% (95% CI: 73% to 86%). Similarly, a calculated threshold of >0.005 ng/mL for troponin I achieved a sensitivity of 65% (95% CI: 56% to 74%) and specificity of 42% (95% CI: 34% to 50%).

Flores et al. [29] prospectively recruited 127 consecutive patients with clinically suspected acute PE admitted to an emergency department at a Spanish university hospital. Blood samples for analyses of the index test tissue plasminogen activator (tPA) were obtained at enrolment and analyzed using enzyme-linked immunosorbent assay (ELISA) (TintElize tPA, Biopool, Sweden). Investigators used the Wells score to categorize patients into a low-, intermediate- or high pretest probability of acute PE, of which, the patients with intermediate and high probability had treatment with either unfractionated or low-molecular-weight heparin, initiated prior to the performance of radiological procedures. A predefined diagnostic threshold for tPA was set to >8.5 ng/mL. The acute PE validation algorithm was complex and included several reference standards: 122 patients received a CTPA, 4 patients received a V/Q scan due to renal insufficiency or contrast allergy, and 1 patient received a necropsy. Furthermore, lower limb ultrasound was performed on two patients who had inconclusive V/Q scans and on three patients who had negative CTPA combined with a high Wells score. The acute PE prevalence was 32%. The sensitivity and specificity of tPA were 95% (95% CI: 83% to 99%) and 36% (95% CI: 26% to 47%), respectively.

Huang et al. [30] included 145 patients with suspected pulmonary embolism, who had been admitted to a hospital in China. An automatic cell counter (SYSMEX XS-1000i; SYSMEX Corporation, Kobe, Japan) was used to determine MPV in blood samples drawn upon admission. All patients underwent CTPA as the reference standard, and the acute PE prevalence was 48%. All test results were collected retrospectively. A diagnostic threshold of >8.45 fL was reported to achieve the highest sensitivity of 89% (95% CI: 79% to 95%) and a specificity of 51% (95% CI: 39% to 62%) in this study population.

Kalkan et al. [31] prospectively included 90 patients with a high clinical probability of acute PE (i.e., patients with a Wells score of ≥7 and/or positive D-dimer test), who had been admitted to an emergency department in a Turkish hospital. Blood samples were drawn early after admission, and the investigators determined NT-proBNP and troponin I concentrations using a chemiluminescent microparticle immunoassay with an ADVIA Centaur kit (Henkestrasse 127 D-91052 Erlangen, Germany). Copeptin levels were measured with a commercially available ELISA kit (Human copeptin ELISA kit, Catalogue No: CK-E90208, Hangzhou East Biopharm CO, Hangzhou, China). All patients underwent CTPA as the reference standard test and had an acute PE prevalence of 52%. For copeptin, a threshold of >4.84 ng/mL was calculated as achieving the highest sensitivity of 68% (95% CI: 53% to 81%) and specificity of 84% (95% CI: 69% to 93%). A NT-proBNP threshold of >247.4 ng/L was calculated to achieve the highest sensitivity of 74% (95% CI: 60% to 86%) and a specificity of 81% (95% CI: 67% to 92%). For troponin I, a threshold of >0.065 ng/mL was calculated to achieve the highest sensitivity and specificity values of 64% (95% CI: 49% to 77%) and 77% (95% CI: 61% to 88%), respectively.

## 4. Discussion

### 4.1. Summary of Main Findings

In this systematic review, estimates of diagnostic accuracy were presented for eight different blood biomarkers to detect or rule out acute PE as a stand-alone test in an emergency department setting. These eight biomarkers were copeptin, MPV, NT-proBNP, PDW, platelet count, RDW, tPA, and troponin I. Only three biomarkers were reported more than once, namely MPV, NT-proBNP, and troponin I, which were reported in two studies. All studies used CTPA as the sole reference standard, except for one study, which used a combination of CTPA, V/Q-scan, and necropsy. Due to the low number of studies for each biomarker, a meta-analysis was not performed. None of the biomarkers had combined high sensitivity and specificity. Only RDW and MPV had a specificity of ≥90% (each reported by one study) [26,27]. However, the specificity of MPV was not consistent in another study (51%) [30], while the specificity of RDW was not investigated in any other study. Selection bias was a major methodological concern in the included studies. Moreover, information on the conduct and interpretation of the reference standard tests, the time interval between the index and reference tests, and the patient flow throughout the studies were inadequately reported. Therefore, the validity of the diagnostic accuracy estimates was uncertain for all eight different biomarkers.

### 4.2. Strengths and Limitations

A major strength of this systematic review was the broad and comprehensive search strategy. Other methodological strengths included a transparent review process, duplicate and independent screening of all records, data extraction using a piloted, standardized form, and risk of bias assessments which were guided by a recognized and structured framework. Moreover, we applied eligibility criteria in order to identify studies reflecting a specific diagnostic setting seen in clinical practice, which could provide valid estimates of diagnostic accuracy [32].

This review has some limitations that merit attention. The patient samples in the included studies were generally younger and healthier than the typical patient population with suspected acute PE reflected in routine clinical practice, while the prevalence of confirmed acute PE by CTPA was higher (32–52%) compared to those commonly observed [10,11,12,13]. These discrepancies between the study samples and the target population indicate that the selected study populations and our eligibility criteria regarding study populations could have been more clearly defined to ensure the inclusion of only clinically relevant populations. Other limitations are the small number of studies investigating the same biomarker, which precluded the pooling of data for a meta-analysis. Unfortunately, none of the eight corresponding authors that we tried to contact responded. However, this would not have changed the decision to refrain from pooling the results for a meta-analysis [33,34,35,36,37,38,39,40].

A major methodological concern in five of six studies [26,27,28,30,31], and a likely contributor to the discrepancy between the patient samples and target population, was the use of inappropriate exclusion criteria. The studies excluded patients with prevalent diseases (e.g., acute coronary syndrome, any hematological disease, diabetes mellitus, COPD, and chronic inflammatory diseases) and patients with common differential diagnoses of acute PE (e.g., myocardial infarction, acute heart failure, and pneumonia), which threaten the external validity of the findings of the study. It may also result in misleading estimates of diagnostic accuracy, thereby threatening the internal validity [21,24,41,42]. Other limitations related to the included studies were the small sample sizes, as well as insufficient reporting of (i) conduct and interpretation of the reference tests, (ii) the time interval between the index (biomarker) and reference standard (CTPA) tests, and (iii) patient flow throughout the studies.

### 4.3. Implications for Practice

None of the eight biomarkers identified in this review had a combined high sensitivity and specificity for acute PE. In the context of searching for blood biomarkers with the potential to improve the current diagnostic algorithm (i.e., reduce the number of patients requiring CTPA), specificity represented the most interesting estimate since the D-dimer already had a high sensitivity of ≥95% [8]. Thus, combining a highly specific blood test with D-dimer could have implications for clinical practice, such as improving the identification of patients requiring CTPA. RDW (by Celik et al. [26]) and MPV (reported by Çevik et al. [27]) had high specificities of 93% and 90%, receptively. However, the reported specificity of MPV was not consistent, as demonstrated by Huang et al. who reported a specificity for MPV of only 51% [30]. Furthermore, the generalizability of the study results by Celik et al. [26] and Çevik et al. [27] remain uncertain due to the data-driven selection of their cut-off values using receiver operating curves (ROCs) that maximize the performance of the test. This approach often increases the chance of over-optimistic diagnostic estimates, and consequently, the results require validation in an external study population to assess whether they are reliable in clinical practice [43,44].

### 4.4. Implications for Future Research

The studies included in this review had a high risk of bias in several important domains. Therefore, well-designed studies are needed to investigate whether blood biomarkers, other than D-dimer, have the potential to improve the diagnostic algorithm for acute PE diagnoses in clinical practice, i.e., to rule out PE in a larger proportion of patients without the need for further referral to CTPA. Future studies on the diagnostic accuracy of blood biomarkers must be comprehensive enough to enable readers to scrutinize the quality of the study and to preserve the spectrum of patients by enrolling study participants in an unselected manner. The former suggestion can be achieved by using the recommended reporting items of the Standards for Reporting Diagnostic Accuracy Studies (STARD) checklist [45]. Adequate study reporting will also help investigators to clearly define the study setting (i.e., use of pretest probability assessment, patient flow, sequence of the index, and reference tests) for biomarker testing. A well-defined study setting, which resembles everyday clinical practice, will increase the reliability and generalizability of the results, and ease the interpretation of pooled estimates of sensitivity and specificity made by future systematic reviews on diagnostic test accuracy. Only one of the studies included in our review used the STARD checklist [28].

Preservation of the total spectrum of patients referred to hospitals with suspected acute PE was especially threatened in the majority of the studies [26,27,28,30,31] owing to patients with common differential diagnoses of acute PE being omitted. This selection creates a healthier group of patients free of acute PE who have a lower probability of producing false positive test results, thereby leading to the falsely high specificity of the diagnostic test [24,41]. For instance, this type of patient selection occurred in the studies by Celik et al. [26] and Çevik et al. [27], further enhancing the possibility of an overestimation in the specificity of RDW and MPV, respectively. In studies on acute PE diagnostics, where diagnostic tests with high specificity were particularly warranted, this type of selection bias is important to prevent. Thus, inappropriate exclusion criteria for the study population must be avoided. Furthermore, it is crucial to avoid sampling patients with and without acute PE from two different source populations. During the eligibility assessment process, we excluded 32 2-gate case–control studies from this review, demonstrating that this design dominates diagnostic research. Importantly, the sampling procedure of this design is known to be one of the largest contributing factors to inaccurate diagnostic estimates [21,46]. Often, mild cases of acute PE that are difficult to diagnose are removed from the case group, while healthy volunteers often constitute the control group. Such sampling procedures will remove the patients with a higher probability of producing false negative and false positive test results, respectively, thereby leading to an overestimation of both sensitivity and specificity [21,46,47]. A control group consisting of patients with the same non-PE disease, such as only myocardial infarction patients, will also influence the diagnostic estimates, and either under- or overestimate the diagnostic accuracy of the blood biomarker depending on the disease type [47]. In addition, several of the acknowledged cost-effectiveness benefits of the case–control design in etiological research is not applicable to diagnostic research [21]. Therefore, a two-gate case–control design should be avoided if the purpose is to investigate the clinical and diagnostic potential of a new blood biomarker.

## 5. Conclusions

This systematic review reports the diagnostic accuracy of eight different blood biomarkers to detect or rule out acute PE using CPTA as the reference standard. The investigated biomarkers were copeptin, MPV, NT-proBNP, PDW, platelet count, RDW, tPA, and troponin I. The modest diagnostic performance and the uncertain validity of the included studies suggest that none of the detected blood biomarkers have the potential to improve the current diagnostic algorithm of acute PE in terms of reducing the use of CTPA. Thus, high-quality studies for the discovery of novel blood biomarkers with acceptable diagnostic accuracy to confirm or rule out acute PE are warranted.

## Figures and Tables

**Figure 1 diagnostics-13-02301-f001:**
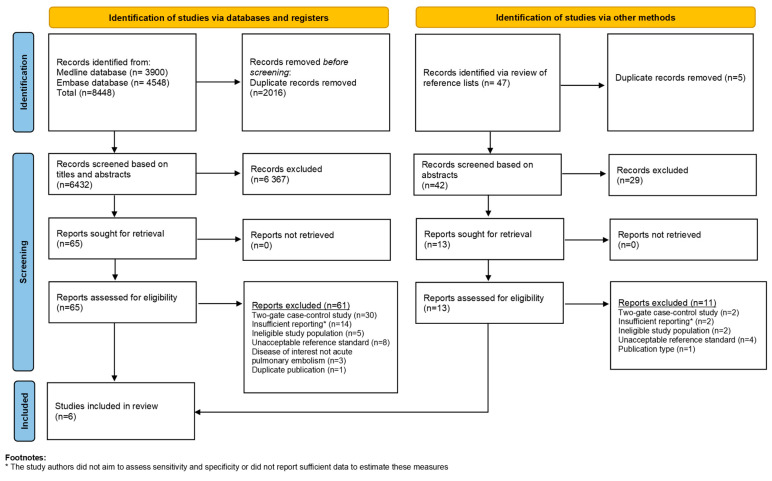
Preferred reporting items for systematic reviews and a meta-analysis flowchart.

**Figure 2 diagnostics-13-02301-f002:**
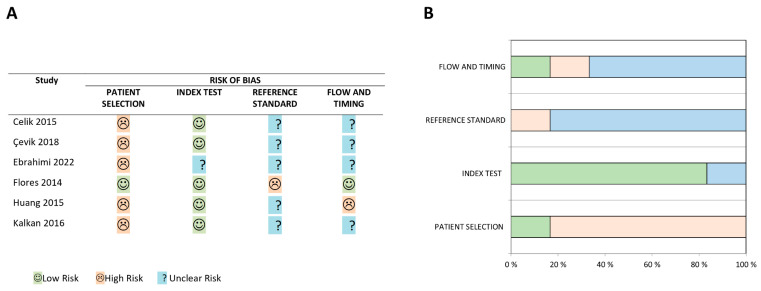
Risk of bias tabular (**A**) [26,27,28,29,30,31] and diagram (**B**) summary: review of authors’ judgments on each bias domain in the quality assessment of diagnostic accuracy studies-2 revised tool [24] for each included study.

**Figure 3 diagnostics-13-02301-f003:**
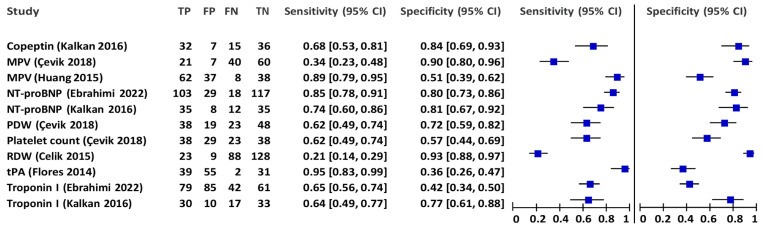
Numerical and graphically (paired forest plots) presentation of the biomarkers’ sensitivity, specificity, and 95% confidence intervals [26,27,28,29,30,31].

**Table 1 diagnostics-13-02301-t001:** Characteristics of included studies, index tests, and reference standards.

First Author, Publication Year, and Country	Timeframe	Population and Clinical Setting	Total Study Size (n), Prevalence of Acute PE (%)	Mean Age, Proportion of Males (%)	Index Test(s) (Threshold)	Reference Standard
Celik et al. 2015, Turkey [26]	January 2011 to May 2013	Patients with suspected acute PE and admitted to hospital (department not specified)	248, 45%	PE+: 59 ± 16 years, 45% PE−: 62 ± 15 years, 46%	Red cell distribution width (<18.9%)	CTPA
Çevik et al. 2018, Turkey [27]	March 2013 to October 2015	Patients with suspected acute PE and admitted to emergency department	128, 48%	PE+: 64 ± 17 years, not reported, PE−: 65 ± 17 years, not reported	Mean platelet volume (≤9 fL) Platelet distribution width (>12.8 fL) Platelet count (≤254 × 10^3^ cells per mL)	CTPA
Ebrahimi et al. 2022, Iran [28]	January 2017 to January 2018	Patients with suspected acute PE admitted to emergency department. PE was suspected when patients had sudden shortness of breath and chest pain and positive D-dimer test result	267, 45%	Only reported for the total study sample, 68 ± 12 years, 60%	NT-proBNP (>100 pg/mL) Troponin I (>0.005 ng/mL)	CTPA
Flores et al. 2014, Spain [29]	September 2008 to October 2009	Patients with suspected acute PE admitted to emergency department. Pretest clinical probability was assessed with Wells score. All patients regardless of Wells score were included	127, 32%	PE+: 62 ± 19 years, 51%, PE−: 50 ± 18 years, 38%	Tissue plasminogen activator (>8.5 ng/mL)	CTPA * V/Q-scan † Necroscopy ‡
Huang et al. 2015, China [30]	September 2009 to January 2014	Patients with suspected acute PE admitted to hospital (department not specified)	145, 48%	PE+: 60 ± 14 years, 46%, PE−: 57 ± 16 years, 57%	Mean platelet volume (>8.45 fL)	CTPA
Kalkan et al. 2016, Turkey [31]	January 2014 to February 2015	Patients with suspected acute PE admitted to emergency department. Pretest clinical probability assessed with Wells score and D-dimer measures. Patients with Wells score ≥ 7 or < 7 and positive D-dimer were included	90, 52%	PE+: 57 ± 16 years, 53%, PE−: 58 ± 16 years, 46%	Copeptin (>4.84 ng/mL) NT-proBNP (>247.4 ng/L) Troponin I (>0.065 ng/mL)	CTPA

Age is reported as mean ± standard deviation. Abbreviations: CTPA: computed tomography pulmonary angiography; NT-proBNP: N-terminal prohormone of brain natriuretic peptide; PE: pulmonary embolism; V/Q-scan: lung ventilation-perfusion scan. PE+ and PE− indicate patients with radiologically confirmed or excluded acute pulmonary embolism, respectively. * 96% of patients received CTPA as reference standard. † 3% of patients received V/Q-scan as reference standard because of renal insufficiency or contrast allergy. ‡ 1% of patients received necroscopy as reference standard because the patient died during diagnostic workup.

**Table 2 diagnostics-13-02301-t002:** Estimates of diagnostic test accuracy for blood biomarkers of acute pulmonary.

Index Test (Biomarker)	TP	FP	FN	TN	Total Study Sample	Sensitivity (95% CIs)	Specificity (95% CIs)	Prevalence of Acute PE	Positive Predictive Value	Negative Predictive Value	Overall Percent Agreement *
Copeptin	32	7	15	36	90	68% (53–81%)	84% (69–93%)	52%	82%	71%	76%
MPV †	21	7	40	60	128	34% (23–48%)	90% (80–96%)	48%	75%	60%	63%
MPV ‡	62	37	8	38	145	89% (79–95%)	51% (39–62%)	48%	63%	83%	69%
NT-proBNP §	103	29	18	117	267	85% (78–91%)	80% (73–86%)	45%	78%	87%	82%
NT-proBNP ¶	35	8	12	35	90	74% (60–86%)	81% (67–92%)	52%	81%	74%	78%
PDW	38	19	23	48	128	62% (49–74%)	72% (59–82%)	48%	67%	68%	67%
Platelet count	38	29	23	38	128	62% (49–74%)	57% (44–69%)	48%	57%	62%	59%
RDW	23	9	88	128	248	21% (14–29%)	93% (88–97%)	45%	72%	59%	61%
tPA	39	55	2	31	127	95% (83–99%)	36% (26–47%)	32%	41%	94%	55%
Troponin I §	79	85	42	61	267	65% (56–74%)	42% (34–50%)	45%	48%	59%	52%
Troponin I ¶	30	10	17	33	90	64% (49–77%)	77% (61–88%)	52%	75%	66%	70%

Abbreviations: CI: confidence interval; MPV: mean platelet volume; NT-proBNP: N-terminal pro-brain natriuretic peptide; PDW: platelet distribution width; PE: pulmonary embolism; RDW: red blood cell distribution width; tPA: tissue plasminogen activator. TP: true positive test results; FP: false positive test results; FN: false negative test results; TN: true negative test results. * (TP + TN)/(TP + TN + FP + FN). The proportion of patients in the study sample was correctly classified as diseased or non-diseased by the index test. †: From the study by Çevik et al. [27]. ‡: From the study by Huang et al. [30]. §: From the study by Ebrahimi et al. [28]. ¶: from the study by Kalkan et al. [31].

## Data Availability

The data presented in this study are available on request from the corresponding author.

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
