# Peer review of "Diagnostic Blood Biomarkers for Acute Pulmonary Embolism: A Systematic Review"

_diagnostics, 2023, doi:10.3390/diagnostics13132301_

Round 1
Reviewer 1 Report
Based on the comments to the authors written below, I would like to recommend the minor revision of the paper.
Please, do not consider this recommendation badly – I sincerely and highly appreciate the effort of the authors to provide a new overview on the diagnosis of acute PE with life-threatening consequences and from my point of view, each novel information is very important for the clinical practice.
Content suggestions:
1. In the effort to include more studies in the review, I would suggest the authors to search the literature also using the terms “CTPA” or “V/Q-scan” to reach the higher number of the studies and more data to exclude these diagnostic procedures from the algorithm of the evaluation of acute PE.
2. Why the authors did not find antithrombin to detect the consumption of one of the natural anticoagulants that is also possible to be substituted to improve the patient´s clinical condition in the case of acute PE ?
3. I would like to kindly ask the authors why they did not assess fibrinogen, D-dimer/fibrinogen ratio (DFR), thrombin generation and further markers of the fibrinolytic processes (plasminogen activator inhibitor 1 (PAI-1), thrombin-activatable fibrinolysis inhibitor (TAFI), alpha-2-antiplasmin), “classical“ BNP, troponin T, lactate, heart-type fatty acid-binding protein (H-FABP), lactate, serum creatinine levels, neutrophil gelatinase-associated lipocalin, cystatin C, hyponatraemia, vasopressin, soluble P-selectin or newly-reported markers, such as tumor necrosis factor-alpha (TNF-α), high mobility histone 1 (HMGB1), plasma signal peptide-complement C1r/C1s, Uegf and Bmp1-epidermal growth factor domain-containing protein 1 (SCUBE-1) ? Was there a lack of adequate data ?
4. Despite the described limitations of the search strategy, can the authors compare the timing / dynamics of the usefulness of the blood sampling of such biomarkers in relation to the time after the first symptoms in the diagnostic algorithm ?
From my point of view, after the addition of the response to these comments, the manuscript can be published. Thus, I recommend the minor revision of the paper.
Reviewer 2 Report
Thank you for the possibility to review the manuscript titled: “Diagnostic Blood Biomarkers for Acute Pulmonary Embolism: A Systematic Review”. The manuscript is well written and easy to read. The only major recommendations is to add more information for a practical approach. Please supplement the manuscript with a table with some of the possible cut-off values that can be used in clinical practice. Also please discuss the possibility to use nomograms or combination of biomarkers to improve diagnostic possibilities.
Please take the recommendations in the spirit of improving the quality of the submission.
